# Fractional Bernstein Series Solution of Fractional Diffusion Equations with Error Estimate

Mohammed Hamed Alshbool [1,*] , Osman Isik [2] and Ishak Hashim [3]

1 Department of Applied Mathematics, Abu Dhabi University, Abu Dhabi 59911, UAE
2 Elemantary Mathematics Education Program, Faculty of Education, Mugla Sitki Kocman University, Mugla 48000, Turkey; osmanrasitisik@hotmail.com
3 Department of Mathematical Sciences, Faculty of Science & Technology, Universiti Kebangsaan Malaysia, UKM Bangi 43600, Malaysia; ishak_h@ukm.edu.my
* Correspondence: alshbool.mohammed@gmail.com

**Abstract:** In the present paper, we introduce the fractional Bernstein series solution (FBSS) to solve the fractional diffusion equation, which is a generalization of the classical diffusion equation. The Bernstein polynomial method is a promising one and can be generalized to more complicated problems in fractional partial differential equations. To get the FBSS, we first convert all terms in the problem to matrix forms. Then, the fundamental matrix equation is obtained and thus, the solution is obtained. Two error estimation methods based on a residual correction procedure and the consecutive approximations are incorporated to find the estimate and bound of the absolute error. The perturbation and stability analysis of the method is given. We apply the method to some illustrative examples. The numerical results are compared with the exact solutions and known second-order methods. The outcomes of the numerical examples are very encouraging and show that the FBSS is highly useful in solving fractional partial problems. The results show the accuracy and effectiveness of the method.

**Keywords:** bernstein series; fractional calculus; diffusion equations; error estimate

## 1. Introduction

Fractional derivatives have been used to model many problems in science, e.g., physics [1–3], medicine [4], hydrology [5], biomedical problems [6], dynamics of particles [7] and applied sciences [8]. The linear fractional diffusion equation is considered for scientists and engineers [9]. Fractional space derivatives are used in the modeling of anomalous diffusion. In a diffusion model, replacement of the second derivative by a fractional derivative causes enhanced diffusion, also called superdiffusion [10].

In the present study, we consider the following one-dimensional fractional diffusion equation (FDE),

$$\frac{\partial y(x,t)}{\partial t} = d(x)\frac{\partial^\alpha y(x,t)}{\partial x^\alpha} + g(x,t), \tag{1}$$

on a finite domain $L < x < R$, $1 < \alpha \le 2$ and $d(x) > 0$. We assume the initial condition $c(x, t = 0) = F(x)$ for $L < x < R$ and the boundary conditions $c(x = L, t) = 0$ and $y(x = R, t) = b_R(t)$.

The fractional derivative in Equation (1) is the Caputo fractional derivative of order $\alpha$ [11]. The basic property of the Caputo derivative is as follows.

$$D_*^\alpha c = 0, \quad (c \quad \text{constant}),$$

$$D_*^\alpha x^\beta = \begin{cases} 0, & \text{for} \quad \beta \in N_0 \quad \text{and} \quad \beta < \lceil \alpha \rceil, \\ \frac{\Gamma(\beta+1)}{\Gamma(\beta+1-\alpha)} x^{\beta-\alpha}, & \text{for} \quad \beta \in N_0 \quad \text{and} \quad \beta \ge \lceil \alpha \rceil \quad \text{or} \quad \beta > \lfloor \alpha \rfloor. \end{cases}$$

Several methods have been used to solve the FDE numerically, e.g., Crank–Nicholson method, which is second-order accurate [12]; finite difference method [13]; finite element method [14]; generalized differential transform method [15]; and Galerkin method, which includes the normalized Bernstein polynomials, collocation method [16], the shifted Jacobi tau method [17], and the Chebyshev spectral-tau method [18]. Fractional Taylor vector to solve multi-term fractional differential equations [19].

One effective method to solve the problems is the Bernstein polynomials method (BPM), also called the Bernstein series solution method [20–25]. Positive linear operators of Bernstein with sequence of these operators are established to solve to solve two-variables equations [26]. The operational matrices of the BPM have been used to get the numerical solutions of a class of third-order ordinary differential equations [27]. The multi-stage BPM, a generalization of the standard BPM, was applied to fractional-order stiff systems [28]. The BPM with new modifications was employed to solve fractional differential equations [29]. Moreover, the same method was used to solve some types of ordinary differential equations [30–32]. A special type of the singular Emden-Fowler problems was solved by BPM [33]. Recently, some linear and non-linear systems of ordinary differential equations have been solved by BPM and the accuracy has been improved by a residual correction procedure [34]. A novel error estimation method for the parametric non-intrusive reduced order model based on machine learning is presented in [35].

In this paper, the fractional Bernstein series solution (FBSS) method is introduced and applied to solve Equation (1) numerically. The method comprises the fractional Bernstein polynomials and collocation method. We approximate the exact solution of Equation (1) by $f_\alpha p_{n,n}$

$$f_\alpha p_{n,n}(x,t) = \sum_{i=1}^n \sum_{j=1}^n a_{ij} f_\alpha B_{i,n}(x) B_{j,n}(t),$$

such that $f_\alpha p_{n,n}$ satisfies Equation (1) on the collocation nodes.

This paper is organized as follows. Section 2 presents necessary definitions and theorems. Section 3, the main section, discusses the matrix forms of the solution $f_\alpha p_{n,n}$ with its derivatives. Then, we get the fundamental matrix equation of the FDE. Employing the conditions and applying the Gauss elimination procedure yields the unknown matrix. Thus, we obtain the FBSS. In Section 4, two different error analysis methods are presented to get the upper bound of the absolute error with the corrected FBSS. Moreover, an upper bound obtained by the generalized Taylor theorem is presented. To decide the stability, the perturbation and stability analysis of the method is done in Section 5. Section 6 provides the numerical results to illustrate the FBSS method for different $n$ values. We compare the method with other methods used to solve the problem. The results show that the present method gives accurate solutions and is also efficient. We test the effect of small perturbations to the approximate solutions both theoretically and practically for various values of $n$. Section 7 summarizes the results.

## 2. Preliminaries and Notations

Bernstein polynomials of $n$-th degree are given by the following equation [29].

$$B_{t,n}(x) = \binom{n}{t} \frac{x^t (R-x)^{n-t}}{R^n}, \quad t = 0, 1, 2, \ldots, n \quad x \in [0, R], \tag{2}$$

Similarly, the fractional Bernstein polynomials $B_{k,n}^\alpha(x)$ are constituted by using $x \to x^\alpha$, so Equation (2) becomes

$$B_{t,n}^\alpha(x) = \binom{n}{t} \frac{x^{t\alpha}(R-x^\alpha)^{n-t}}{R^n}, \quad 1 < \alpha < 2, \tag{3}$$
$$t = 0, 1, 2, \ldots, n \quad x \in [0, R],$$

We will use the definition of Caputo's derivative, which is the modification of the definition of Riemann-Liouville. It has some advantages for solvinginitial value problems [36].

**Definition 1** ([37,38]). *The Riemann–Liouville integral operator of order $\alpha > 0$ for $a \geq 0$ is defined as follows*

$$
\begin{aligned}
(J_a^\alpha f)(x) &= \frac{1}{\Gamma(\alpha)} \int_a^x (x-t)^{\alpha-1} f(t) dt, \ x > a, \\
\left(J_a^0 f\right)(x) &= f(x).
\end{aligned}
$$

**Definition 2** ([37,38]). *The Caputo fractional dervative of $f$ of order $\alpha > 0$ for $a \geq 0$ is given as follows,*

$$
(D_a^\alpha f)(x) = \left(J_a^{m-\alpha} f^{(m)}\right)(x) = \frac{1}{\Gamma(m-\alpha)} \int_a^x \frac{f^{(m)}(t)}{(x-t)^{\alpha+1-m}} dt
$$

*for $m-1 < \alpha \leq m, m \in \mathbb{N}, x \geq a$.*

We will use the following multivariate fractional Taylor's theorem [39] to bound the absolute error.

**Theorem 1.** *For a compact and convex domain $D \subset \mathbb{R}^2$, let $D^{k\alpha} f \in C(D)$ for $k = 0, 1, \ldots, m + 1$ where*

$$
\begin{aligned}
D^{k\alpha} f &= D^{k\alpha - n} D^n f, \ n \text{ is the smallest integer exceeding } k\alpha \\
D^n f &= \left(\Delta x \frac{\partial}{\partial x} + \Delta y \frac{\partial}{\partial y}\right)^n f.
\end{aligned}
$$

*If $(x_0, y_0) \in D$, then*

$$
\begin{aligned}
f(x, y) &= \sum_{k=0}^m \frac{D^{k\alpha} f(x_0, y_0)}{\Gamma(k\alpha + 1)} + \frac{D^{(m+1)\alpha} f(\xi, \eta)}{\Gamma((m+1)\alpha + 1)} \\
&= P_m^\alpha(x, t) + R_m^\alpha(\xi, \eta)
\end{aligned}
$$

*where $\xi = x_0 + \theta \Delta x, \eta = y_0 + \theta \Delta y, 0 < \theta < 1$ and*

$$
\begin{aligned}
P_m^\alpha(x, t) &= \sum_{k=0}^m \frac{D^{k\alpha} f(x_0, y_0)}{\Gamma(k\alpha + 1)} \ (\textit{Truncated mult. frac. Taylor series}) \\
R_m^\alpha(\xi, \eta) &= \frac{D^{(m+1)\alpha} f(\xi, \eta)}{\Gamma((m+1)\alpha + 1)} \ (\textit{Remainder term}).
\end{aligned}
\tag{4}
$$

## 3. Numerical Method

Let

$$
f_\alpha p_{n,n}(x, t) = \sum_{i=1}^n \sum_{j=1}^n a_{ij} f_\alpha B_{i,n}(x) B_{j,n}(t)
$$

be the FBSS of Equation (1). Let us find the matrix forms of

$$
f_\alpha p_{n,n}^\alpha = \frac{\partial^\alpha f_\alpha p_{n,n}}{\partial x^\alpha}, \ 1 < \alpha \leq 2 \text{ and } \frac{\partial f_\alpha p_{n,n}}{\partial t}.
$$

First note that $f_\alpha p_{n,n}$ can be written as follows:

$$f_\alpha p_{n,n}(x,t) = f_\alpha \mathbf{B}_n(x) \mathbf{Q}_n(t) \mathbf{A},$$

where

$$f_\alpha \mathbf{B}_n(x) = \begin{bmatrix} B_{0,n}^\alpha(x) & B_{1,n}^\alpha(x) & \cdots & B_{n,n}^\alpha(x) \end{bmatrix},$$

$$\mathbf{Q}_n(t) = \begin{pmatrix} \mathbf{B}_n(t) & 0 & \dots & 0 \\ 0 & \mathbf{B}_n(t) & \dots & 0 \\ \vdots & \vdots & \ddots & \vdots \\ 0 & 0 & \dots & \mathbf{B}_n(t) \end{pmatrix},$$

and

$$\mathbf{A} = \begin{bmatrix} a_{00} & a_{01} \dots a_{0n} & a_{10} & a_{11} \dots a_{1n} \dots a_{n1} & a_{n2} \dots a_{nn} \end{bmatrix}^T.$$

We write $f_\alpha \mathbf{B}_n(x)$ as

$$f_\alpha \mathbf{B}_n(x) = f_\alpha \mathbf{X}(x) \mathbf{D}^{\mathrm{T}},$$

where

$$\mathbf{D} = \begin{pmatrix} d_{00} & d_{01} & \dots & d_{0n} \\ d_{10} & d_{11} & \dots & d_{1n} \\ \vdots & \vdots & \ddots & \vdots \\ d_{n0} & d_{n1} & \dots & d_{nn} \end{pmatrix}, \quad f_\alpha \mathbf{X}(x) = \begin{bmatrix} 1 & x^\alpha & x^{2\alpha} \dots x^{n\alpha} \end{bmatrix},$$

$$d_{ij} = \begin{cases} \frac{(-1)^{j-i}}{R^j} \binom{n}{i} \binom{n-i}{j-i}, & i \le j \\ 0, & i > j \end{cases}.$$

Therefore, we can write $f_\alpha p_{n,n}$ and $f_\alpha p_{n,n}^\alpha$ in the forms

$$f_\alpha p_{n,n}(x,t) = f_\alpha \mathbf{B}_n(x) \mathbf{Q}_n(t) \mathbf{A}, \tag{5}$$
$$f_\alpha p_{n,n}^\alpha(x,t) = f_\alpha \mathbf{B}_n^\alpha(x) \mathbf{Q}_n(t) \mathbf{A}; \tag{6}$$

respectively. As $f_\alpha \mathbf{B}_n^\alpha(x)$ can be written as

$$f_\alpha \mathbf{B}_n^{(\alpha)}(x) = f_\alpha \mathbf{X}^{(\alpha)}(x) \mathbf{D}^{\mathrm{T}}, \tag{7}$$

for $f_\alpha \mathbf{X}^{(\alpha)}(x)$, the relation

$$f_\alpha \mathbf{X}^{(\alpha)}(x) = f_\alpha \mathbf{X}(x) \mathbf{C}(\mathbf{ff}), \tag{8}$$

is obtained where

$$\mathbf{C}(\mathbf{ff}) = \begin{pmatrix} 0 & 0 & 0 & 0 & \cdots & 0 \\ 0 & 0 & 0 & 0 & \cdots & 0 \\ 0 & 0 & \frac{\Gamma(2)}{\Gamma(2-\alpha)} x^{-\alpha} & 0 & \cdots & 0 \\ 0 & 0 & 0 & \frac{\Gamma(3)}{\Gamma(3-\alpha)} x^{-\alpha} & \cdots & 0 \\ \vdots & \vdots & \vdots & \vdots & \ddots & \vdots \\ 0 & 0 & 0 & 0 & \cdots & \frac{\Gamma(n+1)}{\Gamma(n+1-\alpha)} x^{-\alpha} \end{pmatrix}.$$

Substituting Equation (8) into Equation (7) we get

$$f_\alpha \mathbf{B}_n^{(\alpha)}(x) = f_\alpha \mathbf{X}(x)\mathbf{C}(\mathbf{ff})\mathbf{D}^\mathrm{T}. \tag{9}$$

Note that $\mathbf{Q}_n(t)$ can be given as

$$\mathbf{Q}_n(t) = \bar{\mathbf{Y}}_n(t)\bar{\mathbf{D}}, \tag{10}$$

where

$$\bar{\mathbf{Y}}_n(\mathbf{t}) = \begin{pmatrix} \mathbf{Y}(t) & 0 & \ldots & 0 \\ 0 & \mathbf{Y}(t) & \ldots & 0 \\ \vdots & \vdots & \ddots & \vdots \\ 0 & 0 & \ldots & \mathbf{Y}(t) \end{pmatrix}, \quad \mathbf{Y}(t) = \begin{bmatrix} 1 & t & t^2 \ldots t^n \end{bmatrix},$$

$$\bar{\mathbf{D}} = \begin{pmatrix} \mathbf{D}^\mathrm{T} & 0 & \ldots & 0 \\ 0 & \mathbf{D}^\mathrm{T} & \ldots & 0 \\ \vdots & \vdots & \ddots & \vdots \\ 0 & 0 & \ldots & \mathbf{D}^\mathrm{T} \end{pmatrix}.$$

The term $\dfrac{\partial \mathbf{Q}_n(t)}{\partial t}$ can be written as:

$$\frac{\partial \mathbf{Q}_n(t)}{\partial t} = \frac{\partial \bar{\mathbf{Y}}_n(t)}{\partial t}\bar{\mathbf{D}}, \tag{11}$$

where

$$\frac{\partial \bar{\mathbf{Y}}_n(t)}{\partial t} = \bar{\mathbf{Y}}_n(\mathbf{t})\bar{\mathbf{B}}, \tag{12}$$

and

$$\bar{\mathbf{B}} = \begin{pmatrix} \mathbf{B} & 0 & \ldots & 0 \\ 0 & \mathbf{B} & \ldots & 0 \\ \vdots & \vdots & \ddots & \vdots \\ 0 & 0 & \ldots & \mathbf{B} \end{pmatrix},$$

and

$$\mathbf{B} = \begin{pmatrix} 0 & 1 & 0 & 0 & \ldots & 0 \\ 0 & 0 & 1 & 0 & \ldots & 0 \\ 0 & 0 & 0 & 1 & \ldots & 0 \\ \vdots & \vdots & \ldots & \ldots & \ddots & \vdots \\ 0 & 0 & 0 & 0 & \ldots & 1 \\ 0 & 0 & 0 & 0 & \ldots & 0 \end{pmatrix}.$$

Substituting Equation (12) into Equation (11) we get

$$\frac{\partial \mathbf{Q}_n(t)}{\partial t} = \bar{\mathbf{Y}}(\mathbf{t})\bar{\mathbf{B}}\bar{\mathbf{D}}. \tag{13}$$

Substituting Equations (9) and (13) into Equation (5) yields the matrix forms of $f_\alpha p_{n,n}(x,t)$, $f_\alpha p_{n,n}^\alpha(x,t)$ and $\dfrac{\partial f_\alpha p_{n,n}}{\partial t}$ as

$$f_\alpha p_{n,n}(x,t) = f_\alpha \mathbf{X}(x)\mathbf{D}^{\mathrm{T}}\bar{\mathbf{Y}}(t)\bar{\mathbf{D}}\mathbf{A}, \tag{14}$$

$$f_\alpha p_{n,n}^\alpha(x,t) = f_\alpha \mathbf{X}(x)\mathbf{C}(\mathbf{ff})\mathbf{D}^{\mathrm{T}}\bar{\mathbf{Y}}(t)\bar{\mathbf{D}}\mathbf{A}, \tag{15}$$

$$\frac{\partial f_\alpha p_{n,n}}{\partial t} = f_\alpha \mathbf{X}(x)\mathbf{D}^{\mathrm{T}}\bar{\mathbf{Y}}(t)\bar{\mathbf{B}}\bar{\mathbf{D}}\mathbf{A}, \tag{16}$$

respectively. The use of (15) and (16) in (1) gives the fundamental matrix equation

$$[f_\alpha \mathbf{X}(x)\mathbf{D}^{\mathrm{T}}\bar{\mathbf{Y}}(t)\bar{\mathbf{B}}\bar{\mathbf{D}} - d(x)f_\alpha \mathbf{X}(x)\mathbf{C}(\mathbf{ff})\mathbf{D}^{\mathrm{T}}\bar{\mathbf{Y}}(t)\bar{\mathbf{D}}]\mathbf{A} = g(x,t). \tag{17}$$

Inserting the collocation points $\{(x_i, y_j) : 0 \leq i, j \leq n\}$ in Equation (17) yields the system

$$\mathbf{WA} = \mathbf{G}, \tag{18}$$

where $m$-th row of $\mathbf{W}$ is obtained from $\{(x_k, y_I), k = [|\frac{m}{n+1}|], I = m - k(n+1) - 1$ and

$$[\mathbf{G}]_{1m} = g(x_k, t_I), \quad r = \left[\left|\frac{m}{n+1}\right|\right], \quad I = m - k(n+1) - 1.$$

For the conditions, we first put $t = 0$, $x = L$, and $x = R$ into Equation (14), respectively. Then, we obtain the following matrix relations by substituting the collocation nodes,

$$\begin{aligned}
\mathbf{C_1A} &= \mathbf{G_1}, \tag{19}\\
\mathbf{C_2A} &= \mathbf{G_2},\\
\mathbf{C_3A} &= \mathbf{G_3},
\end{aligned}$$

where

$$\begin{aligned}
[\mathbf{C_1}]_{1,i} &= \mathbf{X}(x_i)\mathbf{D}^{\mathrm{T}}\bar{\mathbf{Y}}(0)\bar{\mathbf{D}},\\
[\mathbf{C_2}]_{1,i} &= \mathbf{X}(L)\mathbf{D}^{\mathrm{T}}\bar{\mathbf{Y}}(t_i)\bar{\mathbf{D}},\\
[\mathbf{C_3}]_{1,i} &= \mathbf{X}(R)\mathbf{D}^{\mathrm{T}}\bar{\mathbf{Y}}(t_i)\bar{\mathbf{D}},\\
[\mathbf{G_1}]_{1,i} &= F(x_i),\\
[\mathbf{G_2}]_{1,i} &= 0,\\
[\mathbf{G_3}]_{1,i} &= b_R(t_i).
\end{aligned}$$

Combining $[\mathbf{W}, \mathbf{G}]$ and $[\mathbf{C_1}, \mathbf{G_1}]$, and $[\mathbf{C_2}, \mathbf{G_2}]$ and $[\mathbf{C_3}, \mathbf{G_3}]$, we obtain a new system $[\tilde{\mathbf{W}}, \tilde{\mathbf{G}}]$:

$$[\tilde{\mathbf{W}}, \tilde{\mathbf{G}}] = \begin{pmatrix} \mathbf{W} & , & \mathbf{G} \\ \mathbf{C_1} & , & \mathbf{G_1} \\ \mathbf{C_2} & , & \mathbf{G_2} \\ \mathbf{C_3} & , & \mathbf{G_3} \end{pmatrix}.$$

Applying the Gauss elimination method to the augmented matrix $[\tilde{\mathbf{W}}, \tilde{\mathbf{G}}]$ and deleting the zero rows gives the system $[\bar{\mathbf{W}}, \bar{\mathbf{G}}]$, where $\bar{\mathbf{W}}$ is square matrix and has full rank. Then, the unknown matrix $\mathbf{A}$ can be obtained as

$$\mathbf{A} = \bar{\mathbf{W}}^{-1}\bar{\mathbf{G}}. \tag{20}$$

## 4. Error Analysis

In this section, first an upper bound for the absolute error is given. Then, we give two error estimation methods that can be applied easily and are useful practically. The first

one is the residual correction procedure. The second one is specifying the consecutive approximations which is similar to the error analysis of the RK4 method.

**Theorem 2.** *Let $f_\alpha p_{n,n}$ and $y$ be the FBSS and the exact solution of Equation (1), respectively. By using the above notations, the absolute error is bounded as follows,*

$$|y(x,t) - f_\alpha p_{n,n}(x,t)| \leq |R_n^\alpha(x,t)| + |P_n^\alpha(x,t) - f_\alpha p_{n,n}(x,t)|$$

*provided that $D^{k\alpha} f \in C(D)$, where $D$ is a rectangular domain contains the collocation nodes.*

**Proof.** Adding and subtracting the term $P_n^\alpha(x,t)$, the two-dimensional truncated generalized Taylor polynomial defined in (4), into the left-hand side and applying triangle inequality yields the desired result. □

Let $R$ be the function defined as

$$R(x,t) := \frac{\partial f_\alpha p_{n,n}(x,t)}{\partial t} - d(x)\frac{\partial^\alpha f_\alpha p_{n,n}(x,t)}{\partial x^\alpha}.$$

Adding the term $R$ into both sides of Equation (1) yields the following fractional differential equation for the absolute error

$$\frac{\partial e_{n,n}(x,t)}{\partial t} = d(x)\frac{\partial^\alpha e_{n,n}(x,t)}{\partial x^\alpha} - \frac{\partial f_\alpha p_{n,n}(x,t)}{\partial t} + d(x)\frac{\partial^\alpha f_\alpha p_{n,n}(x,t)}{\partial x^\alpha} + g(x,t), \qquad (21)$$

where $e_{n,n} = y - f_\alpha p_{n,n}$. The initial and boundary conditions for the problem are converted to

$$\begin{aligned} e_{n,n}(x,0) &= 0, \\ e_{n,n}(L,t) &= e_{n,n}(R,t) = 0. \end{aligned} \qquad (22)$$

We get an approximate solution, $e_{n,n}^{m,m}$, for the absolute error by applying the method to Equation (21) with the conditions (22) on the nodes $\{(x_i, t_j) : 0 \leq i, j \leq m\} \subset \Omega$. Then, the absolute error $e_{n,n}$ can be estimated by $e_{n,n}^{m,m}$ provided that $\|e_{n,n} - e_{n,n}^{m,m}\| < \varepsilon$ is small.

**Corollary 1.** *Let $f_\alpha p_{n,n}$ be the FBSS. Then, $f_\alpha p_{n,n} + e_{n,n}^{m,m}$ is another approximate solution, corrected FBSS, of Equation (1) and its error function is $e_{n,n} - e_{n,n}^{m,m}$. Moreover, if*

$$\|e_{n,n} - e_{n,n}^{m,m}\| < \|y - e_{n,n}\|,$$

*then $f_\alpha p_{n,n} + e_{n,n}^{m,m}$ is a better approximation than $p_{n,n}$ in any given norm $\|\cdot\|$.*

Let $f_\alpha p_{n,n}$ and $f_\alpha p_{s,s}$ be any two FBSS of Equation (1), $y$ be the exact solution of Equation (1). Then, by using the triangle inequality, we find the following inequality,

$$\|e_{n,n}\| - \|e_{m,m}\| = \|y - f_\alpha p_{n,n}\| - \|y - f_\alpha p_{m,m}\| \leq \|f_\alpha p_{n,n} - f_\alpha p_{m,m}\|.$$

If $\|e_{m,m}\| < \|e_{n,n}\|$, then we can write as follows

$$\|e_{n,n}\| = C\|e_{m,m}\|, \quad C > 1.$$

Therefore, we can bound the error as

$$\begin{aligned} \|e_{n,n}\| - \|e_{m,m}\| &= (C-1)\|e_{m,m}\| \leq \|f_\alpha p_{m,m} - f_\alpha p_{n,n}\| \text{ or} \\ \|e_{n,n}\| - \|e_{m,m}\| &= (1 - \frac{1}{C})\|e_{n,n}\| \leq \|f_\alpha p_{m,m} - f_\alpha p_{n,n}\|. \end{aligned}$$

Therefore, we can bound the error by

$$\frac{1}{(C-1)}\|f_\alpha p_{m,m} - f_\alpha p_{n,n}\| \text{ or } \frac{C}{(C-1)}\|f_\alpha p_{m,m} - f_\alpha p_{n,n}\|. \tag{23}$$

Then, we can bound the error $\|e_{m,m}\|$ well in case of $C \geq 2$, even when the exact solution is not known. A similar argument may be proposed when the error sequence is decreasing (or increasing). Thus, in case of a decreasing (or increasing) error sequence, one of the following bounds is satisfied

$$\|e_{n,n}\| \leq \frac{1}{C-1}\|f_\alpha p_{n+1,n+1} - f_\alpha p_{n,n}\|$$

or

$$\|e_{n,n}\| \leq \frac{C}{C-1}\|f_\alpha p_{n+1,n+1} - f_\alpha p_{n,n}\|.$$

## 5. Numerical Stability

In this section, we will study the perturbation analysis with stability estimation of the linear systems obtained by the FBSS method, which is similar to that in [40], for a given problem. Perturbing the initial or boundary conditions yields the following perturbed solutions, $f_\alpha p_{n,n}^{per}(x,t)$. There occur two cases:

**Case 1:** Only the vector $\bar{\mathbf{G}}$ on the right hand side of (20) is perturbed, i.e.,

$$\bar{\mathbf{W}}\mathbf{A} = \bar{\mathbf{G}} + \Delta\bar{\mathbf{G}}. \tag{24}$$

Let us show the perturbed solution of (24) as $\mathbf{A}^p = \mathbf{A} + \Delta\mathbf{A}$, where $\Delta\mathbf{A}$ is the perturbation of the solution resulting from the perturbations in the initial and boundary conditions. Then, the change in the solution caused by the initial change is bounded as [41]

$$\frac{\|\Delta\mathbf{A}\|}{\|\mathbf{A}\|} \leq \text{cond}(\bar{\mathbf{W}})\frac{\|\Delta\bar{\mathbf{G}}\|}{\|\bar{\mathbf{G}}\|}.$$

**Case 2:** The perturbations might be occurred both $\bar{\mathbf{W}}$ and $\bar{\mathbf{G}}$,

$$(\bar{\mathbf{W}} + \Delta\bar{\mathbf{W}})\mathbf{A} = \bar{\mathbf{G}} + \Delta\bar{\mathbf{G}}. \tag{25}$$

As the same notation in Case 1, the change in $\mathbf{A}$ caused by perturbing the initials is bounded above as [41]

$$\frac{\|\Delta\mathbf{A}\|}{\|\mathbf{A}\|} \leq \frac{\text{cond}(\bar{\mathbf{W}})}{1 - \text{cond}(\bar{\mathbf{W}})\frac{\|\Delta\bar{\mathbf{W}}\|}{\|\bar{\mathbf{W}}\|}}\left(\frac{\|\Delta\bar{\mathbf{W}}\|}{\|\bar{\mathbf{W}}\|} + \frac{\|\Delta\bar{\mathbf{G}}\|}{\|\bar{\mathbf{G}}\|}\right).$$

Thus, for Case 1,

$$\begin{aligned}\left|f_\alpha p_{n,n}(x,t) - f_\alpha^{per} p_{n,n}(x,t)\right| &= |f_\alpha \mathbf{B}_n(x)\mathbf{Q}_n(t)(\mathbf{A} - \mathbf{A}^p)| \tag{26}\\ &\leq \|f_\alpha \mathbf{B}_n(x)\|\|\mathbf{Q}_n(t)\|\|\Delta\mathbf{A}\| \\ &\leq \|f_\alpha \mathbf{B}_n(x)\|\|\mathbf{Q}_n(t)\|\text{cond}(\bar{\mathbf{W}})\frac{\|\Delta\bar{\mathbf{G}}\|\|\mathbf{A}\|}{\|\bar{\mathbf{G}}\|}.\end{aligned}$$

and therefor we can specify the effect of the little changes in the initial and boundary conditions on the FBSS by measuring $\text{cond}(\bar{\mathbf{W}})$. We omit the result for Case 2 by simplicity.

## 6. Numerical Results and Discussion

In this section, three examples are provided to illustrate the properties and effectiveness of the technique.

### 6.1. Example 1

Let us consider the FDE [17]

$$\frac{\partial u(x,t)}{\partial t} = d(x)\frac{\partial^{1.8}u(x,t)}{\partial x^{1.8}} + g(x,t), \tag{27}$$

where $0 < x < 1$ and $\alpha = 1.8$, the diffusion coefficient is

$$d(x) = \Gamma(2.2)\frac{x^{2.8}}{6}$$

and the source function

$$g(x,t) = -(1+x)e^{-t}x^3.$$

The initial condition is

$$u(x,0) = x^3$$

and the boundary conditions are

$$u(0,t) = 0, \quad u(1,t) = e^{-t}, \quad t > 0.$$

The exact solution to this problem is

$$u(x,t) = e^{-t}x^3.$$

By applying the procedure in Section 3, the matrix equation for Equation (27) is found as

$$[\mathbf{X}(x)\mathbf{D}^{\mathsf{T}}\bar{\mathbf{Y}}(t)\bar{\mathbf{B}}\bar{\mathbf{D}} - \Gamma(2.2)\frac{x^{2.8}}{6}\mathbf{X}(x)\mathbf{C}(\mathbf{ff})\mathbf{D}^{\mathsf{T}}\bar{\mathbf{Y}}(t)\bar{\mathbf{D}}]\mathbf{A} = -(1+x)e^{-t}x^3. \tag{28}$$

The collocation points that will be used are the Chebyshev interpolation nodes

$$\{(x_i, y_j) : 0 \le i,j \le n, \quad x_i = \frac{1}{2} + \frac{1}{2}\cos(\frac{2i-1}{2n})\pi, \quad y_j = \frac{1}{2} + \frac{1}{2}\cos(\frac{2j-1}{2n})\pi\}.$$

By substituting the collocation points in Equation (28), we will obtain $\mathbf{W}$ matrix. The conditions matrices for $(x,0) = x^3, u(0,t) = 0, u(1,t) = e^{-t}$ are obtained as

$$p_{n,n}(x_i,0) = \mathbf{X}(x_i)\mathbf{D}^{\mathsf{T}}\bar{\mathbf{Y}}(0)\bar{\mathbf{D}}\mathbf{A} = x_i^3,$$
$$p_{n,n}(0,t_j) = \mathbf{X}(0)\mathbf{D}^{\mathsf{T}}\bar{\mathbf{Y}}(t_j)\bar{\mathbf{D}}\mathbf{A} = 0,$$
$$p_{n,n}(1,t_j) = \mathbf{X}(1)\mathbf{D}^{\mathsf{T}}\bar{\mathbf{Y}}(t_j)\bar{\mathbf{D}}\mathbf{A} = e^{-t_j}, \quad 0 \le i,j,k \le n.$$

Then, $[\tilde{\mathbf{W}}, \tilde{\mathbf{G}}]$ is obtained by combining these matrices. Thus, we obtain the coefficient matrix $\mathbf{A}$. As we perform the method for different $n$ values, we obtain $n$ different FBSS. All calculations are done using the Maple program. The results for $n = 5, 10, n = 15$, and $n = 20$ are given in Table 1. The absolute errors are graphed in Figure 1. Table 2 show the absolute errors between consecutive approximations. The exact and approximate solutions with $n = 15$ are graphed in Figure 2. The values of error functions and the exact solution, at time $t = 1$, are given in Table 3 and graphed in Figure 3. The upper bounds of absolute error at time $t = 1$ are given in Figure 4. A comparison of the method with the numerical methods in [16,18] is made for $t = 1$. The absolute error for $n = 5$ with the errors obtained

by the corrected FBSSs are given in Table 4. We can say that increasing $n$ yields better approximation results and the method provides better results. From Table 2, we deduce that the absolute error $e_{n,n}$ can be estimated approximately by the difference between $e_{n,n}$ and $e_{n+1,n+1}$. From Table 3, we can see the method gives more accurate results than the methods in [16,18] for $t = 1$. From Table 4, increasing $m$ yields more accurate corrected approximate solutions. Table 5 shows the error norms $(L_2, L_\infty)$ resulting with CPU time (in seconds) used in the Maple program to find the numerical solutions for different $n$ values. We provide the stability results for the method for some $n$ values in Table 6. Increasing $n$ gives the condition numbers which are increasing. It can be said that the approximate solutions will be more stable around $n = 10$ by using both the theoretical upper bounds and the numerical results. The method produces approximately $10^5$ as a condition number. For $n \gg 10$, the solutions will be more sensitive to small variations since the number of conditions is large. Thus, around $n = 10$, we can say that the method works well for this problem.

**Table 1.** Comparisons of the absolute errors for different $n$ for Example 1.

| $x$ | $t$ | $|e_{5,5}|$ | $|e_{10,10}|$ | $|e_{15,15}|$ | $|e_{20,20}|$ |
|-----|-----|-------------|---------------|---------------|---------------|
| 0.0 | 0.0 | $8.096 \times 10^{-5}$ | $1.030 \times 10^{-6}$ | $9.528 \times 10^{-8}$ | $1.678 \times 10^{-8}$ |
| 0.2 | 0.2 | $7.314 \times 10^{-5}$ | $2.828 \times 10^{-7}$ | $9.191 \times 10^{-9}$ | $6.885 \times 10^{-9}$ |
| 0.4 | 0.4 | $1.148 \times 10^{-4}$ | $3.347 \times 10^{-7}$ | $2.655 \times 10^{-8}$ | $7.076 \times 10^{-10}$ |
| 0.6 | 0.6 | $5.966 \times 10^{-5}$ | $9.071 \times 10^{-7}$ | $2.046 \times 10^{-9}$ | $5.669 \times 10^{-9}$ |
| 0.8 | 0.8 | $3.015 \times 10^{-5}$ | $8.664 \times 10^{-8}$ | $1.116 \times 10^{-8}$ | $9.309 \times 10^{-8}$ |
| 1.0 | 1.0 | $7.439 \times 10^{-7}$ | $1.744 \times 10^{-10}$ | $1.789 \times 10^{-10}$ | $5.641 \times 10^{-11}$ |

**Table 2.** Upper bounds for the absolute errors for $n = 9$ and $n = 10$ of Example 1.

| $x$ | $t$ | $|y - y_9|$ | $|y - y_{10}|$ | $|y_9 - y_{10}|$ |
|-----|-----|-------------|----------------|------------------|
| 0.0 | 0.0 | $2.138 \times 10^{-6}$ | $1.030 \times 10^{-6}$ | $1.107 \times 10^{-6}$ |
| 0.2 | 0.2 | $6.193 \times 10^{-7}$ | $2.828 \times 10^{-7}$ | $9.022 \times 10^{-6}$ |
| 0.4 | 0.4 | $2.488 \times 10^{-7}$ | $3.347 \times 10^{-7}$ | $2.153 \times 10^{-6}$ |
| 0.6 | 0.6 | $2.278 \times 10^{-6}$ | $9.071 \times 10^{-7}$ | $1.371 \times 10^{-6}$ |
| 0.8 | 0.8 | $1.527 \times 10^{-6}$ | $8.664 \times 10^{-8}$ | $6.661 \times 10^{-6}$ |
| 1.0 | 1.0 | $2.344 \times 10^{-9}$ | $1.744 \times 10^{-10}$ | $2.519 \times 10^{-9}$ |

**Table 3.** Absolute errors on $[0, 1]$, with $n = 5$, $n = 10$ and $n = 15$ for Example 1 at time $t = 1$.

| $x$ | $n = 5$ | $n = 10$ | $n = 15$ | Method in [16] | Method in [18] |
|-----|---------|----------|----------|----------------|----------------|
| 0.1 | $7.883 \times 10^{-7}$ | $2.705 \times 10^{-9}$ | $1.552 \times 10^{-10}$ | $2.7 \times 10^{-8}$ | $1.9 \times 10^{-6}$ |
| 0.2 | $4.005 \times 10^{-5}$ | $5.618 \times 10^{-7}$ | $2.877 \times 10^{-9}$ | $6.5 \times 10^{-7}$ | $1.4 \times 10^{-7}$ |
| 0.3 | $1.017 \times 10^{-4}$ | $3.809 \times 10^{-7}$ | $2.154 \times 10^{-8}$ | $5.1 \times 10^{-7}$ | $2.6 \times 10^{-6}$ |
| 0.4 | $1.080 \times 10^{-4}$ | $6.936 \times 10^{-7}$ | $4.0434 \times 10^{-8}$ | $3.2 \times 10^{-6}$ | $3.8 \times 10^{-6}$ |
| 0.5 | $9.053 \times 10^{-5}$ | $8.913 \times 10^{-7}$ | $3.881 \times 10^{-8}$ | $5.8 \times 10^{-6}$ | $3.4 \times 10^{-6}$ |
| 0.6 | $6.758 \times 10^{-5}$ | $8.504 \times 10^{-7}$ | $2.398 \times 10^{-8}$ | $6.1 \times 10^{-6}$ | $2.0 \times 10^{-6}$ |
| 0.7 | $4.838 \times 10^{-5}$ | $4.054 \times 10^{-7}$ | $5.763 \times 10^{-9}$ | $3.1 \times 10^{-6}$ | $4.5 \times 10^{-7}$ |
| 0.8 | $3.355 \times 10^{-5}$ | $7.251 \times 10^{-7}$ | $1.025 \times 10^{-8}$ | $2.2 \times 10^{-6}$ | $3.3 \times 10^{-7}$ |
| 0.9 | $1.872 \times 10^{-6}$ | $2.869 \times 10^{-7}$ | $1.335 \times 10^{-9}$ | $6.1 \times 10^{-6}$ | $9.7 \times 10^{-8}$ |

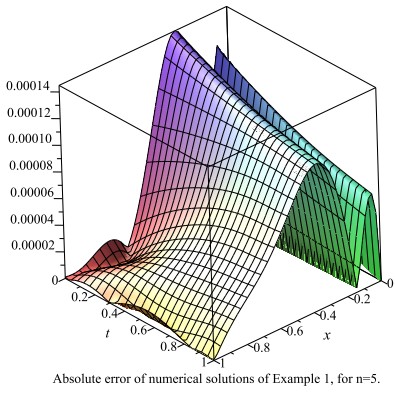

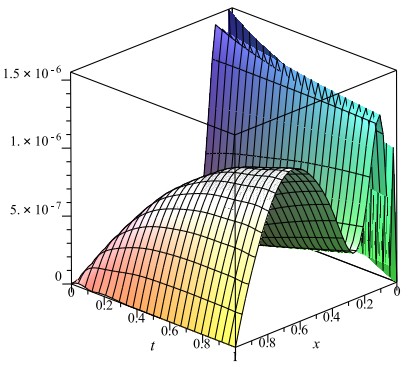

Absolute error of numerical solutions of Example 1, for n=5.

Absolute error of numerical solutions of Example 1, for n=10

Absolute error of numerical solutions of Example 1, for n=15

**Figure 1.** The absolute errors for Example 1: $n = 5$, $n = 10$ and $n = 15$.

**Table 4.** The absolute errors with the corrected FBSSs $n = 5$, and $m = 10, 15, 20$ of Example 1.

| $x$ | $t$ | $e_{5,5}$ | $e_{5,5}^{10,10}$ | $e_{5,5}^{15,15}$ | $e_{5,5}^{20,20}$ |
|---|---|---|---|---|---|
| 0.0 | 0.0 | $8.096 \times 10^{-5}$ | $1.825 \times 10^{-6}$ | $2.621 \times 10^{-7}$ | $4.519 \times 10^{-8}$ |
| 0.2 | 0.2 | $7.314 \times 10^{-5}$ | $8.021 \times 10^{-5}$ | $7.156 \times 10^{-5}$ | $7.191 \times 10^{-6}$ |
| 0.4 | 0.4 | $1.148 \times 10^{-4}$ | $1.084 \times 10^{-4}$ | $1.029 \times 10^{-4}$ | $9.817 \times 10^{-5}$ |
| 0.6 | 0.6 | $5.966 \times 10^{-5}$ | $5.071 \times 10^{-5}$ | $3.681 \times 10^{-5}$ | $2.295 \times 10^{-6}$ |
| 0.8 | 0.8 | $3.015 \times 10^{-5}$ | $2.947 \times 10^{-5}$ | $8.609 \times 10^{-6}$ | $4.658 \times 10^{-6}$ |
| 1.0 | 1.0 | $7.439 \times 10^{-7}$ | $7.224 \times 10^{-7}$ | $1.158 \times 10^{-7}$ | $7.392 \times 10^{-7}$ |

**Table 5.** Error norms $L_2$, $L_\infty$ and CPU for different values of $n$ for Example 1.

| $n$ | $L_2$ | $L_\infty$ | CPU |
|---|---|---|---|
| 3 | $4.403 \times 10^{-2}$ | $2.343 \times 10^{-2}$ | 6.30 |
| 5 | $2.454 \times 10^{-4}$ | $1.148 \times 10^{-4}$ | 10.5 |
| 8 | $4.328 \times 10^{-6}$ | $8.541 \times 10^{-6}$ | 15.5 |
| 10 | $2.582 \times 10^{-6}$ | $1.472 \times 10^{-6}$ | 18.3 |
| 13 | $4.079 \times 10^{-7}$ | $2.186 \times 10^{-7}$ | 90.5 |
| 15 | $1.074 \times 10^{-7}$ | $9.528 \times 10^{-8}$ | 129.7 |
| 18 | $1.181 \times 10^{-7}$ | $7.121 \times 10^{-8}$ | 472.1 |
| 20 | $1.061 \times 10^{-7}$ | $7.067 \times 10^{-8}$ | 750.5 |

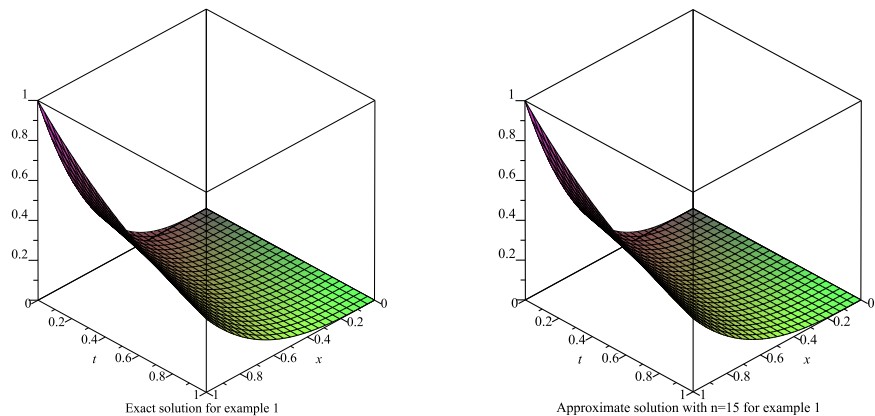

**Figure 2.** The exact solution and FBSSs for Example 1.

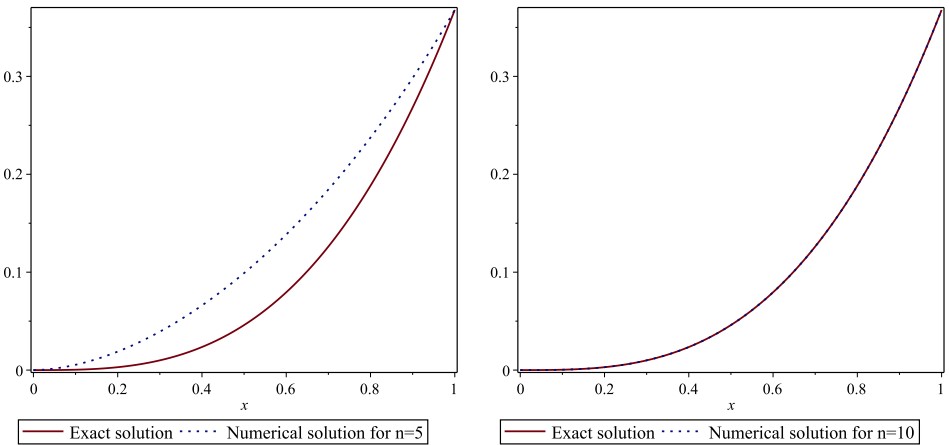

**Figure 3.** FBSSs and the exact solution for Example 1, with $n = 5$ and $n = 10$ at time $t = 1$.

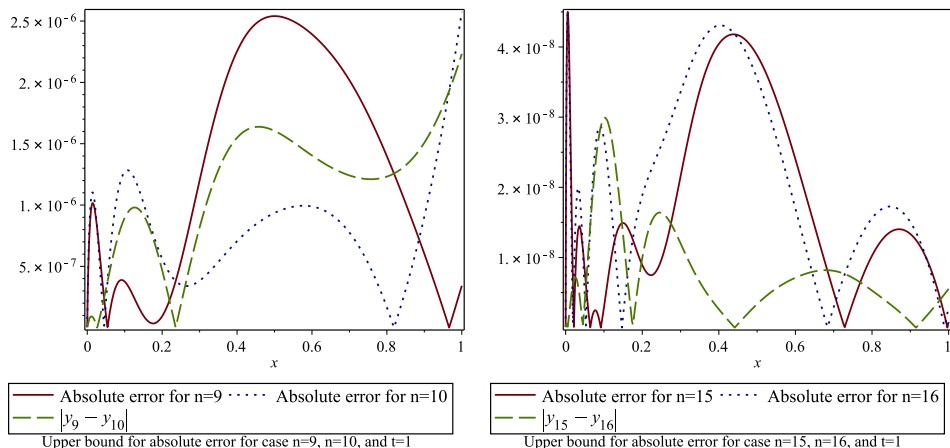

**Figure 4.** Upper bounds of the absolute errors for Example 1, with $n = 9, 10$ and $n = 15, 16$ at time $t = 1$.

**Table 6.** Stability of the system related to the method for different values of $n$ for Example 1.

|  | $n = 5$ | $n = 10$ | $n = 15$ |
|---|---|---|---|
| $\text{cond}(\bar{W})$ | 673.48 | $6.33 \times 10^5$ | $1.68 \times 10^9$ |
| $\|\Delta A\|$ | $1.0253 \times 10^{-16}$ | $1.5394 \times 10^{-15}$ | $4.9009 \times 10^{-9}$ |
| $\|A\|$ | 1.0000 | 1.0000 | 0.9999 |
| $\|\Delta \bar{G}\|$ | $10^{-16}$ | $10^{-16}$ | $10^{-16}$ |
| $\|\bar{G}\|$ | 1.0775 | 2.1642 | 4.1801 |
| Upper Bound obtained by (26) | $2 \times 10^{-10}$ | $5.9987 \times 10^{-7}$ | $3.9567 \times 10^{-2}$ |

*6.2. Example 2*

Let us consider the FDE [17]

$$\frac{\partial u(x,t)}{\partial t} = d(x)\frac{\partial^{1.4} u(x,t)}{\partial x^{1.4}} + g(x,t), \tag{29}$$

where $0 < x < 1$ and $\alpha = 1.4$. The diffusion coefficient is

$$d(x) = \frac{1}{24}\Gamma(5 - 1.4)x^{1.4},$$

the source function

$$g(x,t) = -2e^{-t}x^4.$$

The initial and boundary conditions are as follows, respectively,

$$u(x,0) = x^4$$

$$u(0,t) = 0, \quad u(1,t) = e^{-t}, \quad t > 0.$$

The exact solution of this problem is given by

$$u(x,t) = e^{-t}x^4.$$

We perform the method for $n = 7, 10$ and $n = 15$. The results are given in Table 7. The absolute errors are graphed in Figure 5. A comparison of the method with the numerical method in [17] is given in Table 7. We can say that increasing $n$ yields better approximation results and the method provides better results. The method gives more accurate results than the method in [17]. From Table 8, we conclude that the absolute error $e_{n,n}$ can be bounded approximately by $|e_{n,n} - e_{n+1,n+1}|$. The absolute error for $n = 5$ with the errors obtained by the corrected FBSSs are given in Table 9. The absolute errors for $n = 11$ and $n = 16$ at time $t = 1$ are given in Figure 6. The errors with their upper bounds at $t = 1$ are given in Figure 7. Table 10 shows the error norms $(L_2, L_\infty)$, resulting with CPU time (in seconds) used in the Maple program to find FBSSs for different $n$ values. The method again gives approximately $10^5$ as a condition number. Therefore, around $n = 10$, we can say from Table 11 that the method produces more stable approximations for this problem.

*6.3. Example 3*

Let us consider the fractional convection diffusion equation [42]

$$\frac{\partial u(x,t)}{\partial t} + d(x)\frac{\partial u(x,t)}{\partial x} = c(x)\frac{\partial^2 u(x,t)}{\partial x^2} + g(x,t), \tag{30}$$

where $0 < x < 1$ and. The diffusion coefficient is

$$d(x) = \frac{x}{3}, \quad c(x) = \frac{x^2}{6}.$$

the source function

$$g(x,t) = x^3 \cosh(t).$$

The initial and boundary conditions are as follows, respectively,

$$u(x,0) = 0$$

$$u(0,t) = 0, \quad u(1,t) = \sinh(t).$$

The exact solution of this problem is given by

$$u(x,t) = x^3 \sinh(t).$$

**Table 7.** The absolute errors for different $n$ values with a comparison with the method in [17] for Example 2.

| $x$ | $t$ | $n = 8$ | $n = 10$ | $n = 15$ | Method in [17] |
|-----|-----|---------|----------|----------|----------------|
| 0.1 | 0.1 | $1.421 \times 10^{-7}$ | $7.187 \times 10^{-9}$ | $1.343 \times 10^{-13}$ | $2.7 \times 10^{-6}$ |
| 0.2 | 0.2 | $7.425 \times 10^{-8}$ | $8.404 \times 10^{-9}$ | $6.504 \times 10^{-11}$ | $1.1 \times 10^{-5}$ |
| 0.3 | 0.3 | $3.189 \times 10^{-8}$ | $24.59 \times 10^{-8}$ | $2.747 \times 10^{-10}$ | $1.2 \times 10^{-5}$ |
| 0.4 | 0.4 | $2.147 \times 10^{-7}$ | $1.310 \times 10^{-8}$ | $3.070 \times 10^{-10}$ | $1.2 \times 10^{-5}$ |
| 0.5 | 0.5 | $2.152 \times 10^{-7}$ | $1.253 \times 10^{-8}$ | $5.414 \times 10^{-10}$ | $1.5 \times 10^{-5}$ |
| 0.6 | 0.6 | $3.730 \times 10^{-8}$ | $1.076 \times 10^{-8}$ | $6.843 \times 10^{-10}$ | $1.8 \times 10^{-5}$ |
| 0.7 | 0.7 | $3.411 \times 10^{-7}$ | $1.787 \times 10^{-8}$ | $7.885 \times 10^{-10}$ | $1.8 \times 10^{-5}$ |
| 0.8 | 0.8 | $2.716 \times 10^{-7}$ | $3.792 \times 10^{-8}$ | $6.409 \times 10^{-10}$ | $1.3 \times 10^{-5}$ |
| 0.9 | 0.9 | $1.262 \times 10^{-7}$ | $1.125 \times 10^{-10}$ | $4.291 \times 10^{-12}$ | $6.6 \times 10^{-6}$ |

**Table 8.** Upper bounds of the absolute errors, for $n = 10$, $n = 11$ and Example 2.

| $x$ | $t$ | $|y - y_{10}|$ | $|y - y_{11}|$ | $|y_{10} - y_{11}|$ |
|-----|-----|----------------|----------------|---------------------|
| 0.0 | 0.0 | $3.201 \times 10^{-9}$ | $9.861 \times 10^{-10}$ | $7.911 \times 10^{-10}$ |
| 0.2 | 0.2 | $8.404 \times 10^{-9}$ | $4.976 \times 10^{-9}$ | $2.924 \times 10^{-8}$ |
| 0.4 | 0.4 | $1.310 \times 10^{-8}$ | $2.389 \times 10^{-8}$ | $2.576 \times 10^{-8}$ |
| 0.6 | 0.6 | $1.076 \times 10^{-8}$ | $3.348 \times 10^{-8}$ | $1.682 \times 10^{-7}$ |
| 0.8 | 0.8 | $3.792 \times 10^{-8}$ | $4.166 \times 10^{-8}$ | $1.076 \times 10^{-6}$ |
| 1.0 | 1.0 | $9.600 \times 10^{-10}$ | $7.996 \times 10^{-11}$ | $2.110 \times 10^{-6}$ |

**Table 9.** The absolute errors with the corrected FBSSs for $n = 5$ and $m = 10, 15, 20$ of Example 2.

| $x$ | $t$ | $e_{5,5}$ | $e_{5,5}^{10,10}$ | $e_{5,5}^{15,15}$ | $e_{5,5}^{20,20}$ |
|-----|-----|-----------|-------------------|-------------------|-------------------|
| 0.0 | 0.0 | $3.401 \times 10^{-3}$ | $5.042 \times 10^{-7}$ | $3.603 \times 10^{-7}$ | $6.510 \times 10^{-8}$ |
| 0.2 | 0.2 | $5.411 \times 10^{-4}$ | $2.017 \times 10^{-4}$ | $2.174 \times 10^{-4}$ | $2.009 \times 10^{-5}$ |
| 0.4 | 0.4 | $6.232 \times 10^{-4}$ | $1.646 \times 10^{-5}$ | $1.600 \times 10^{-5}$ | $2.641 \times 10^{-5}$ |
| 0.6 | 0.6 | $1.356 \times 10^{-3}$ | $2.266 \times 10^{-5}$ | $3.024 \times 10^{-5}$ | $4.091 \times 10^{-5}$ |
| 0.8 | 0.8 | $1.201 \times 10^{-3}$ | $2.752 \times 10^{-5}$ | $3.847 \times 10^{-5}$ | $6.655 \times 10^{-5}$ |
| 1.0 | 1.0 | $1.413 \times 10^{-5}$ | $1.143 \times 10^{-6}$ | $2.427 \times 10^{-6}$ | $4.372 \times 10^{-7}$ |

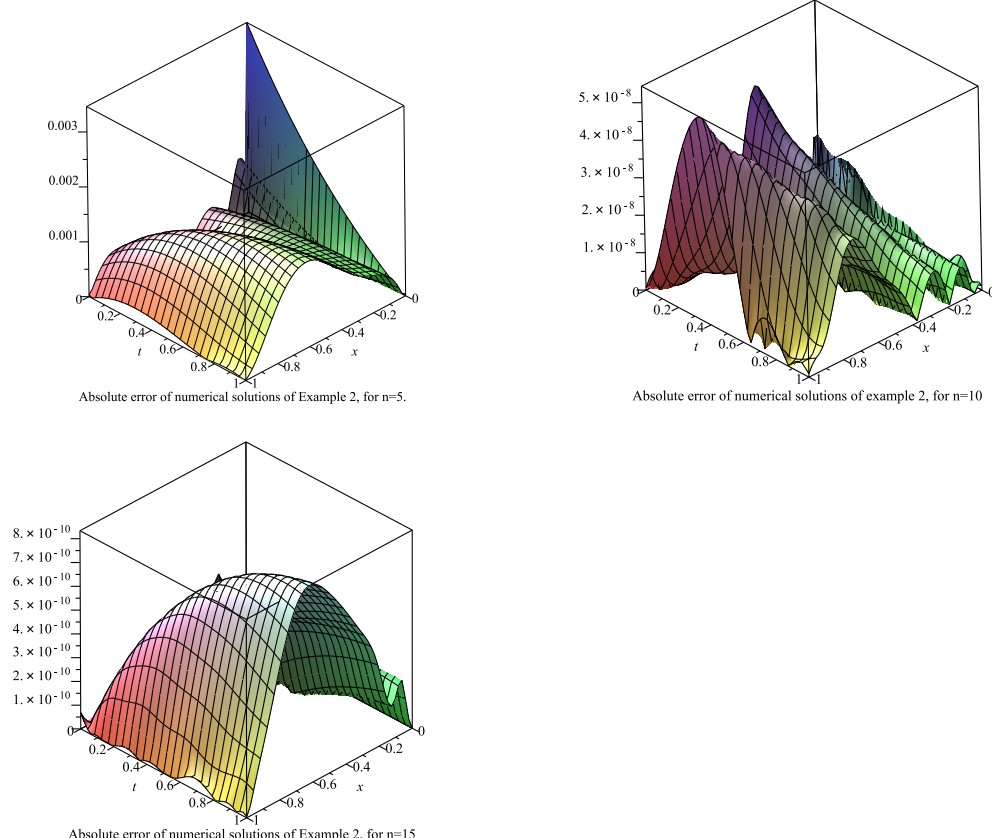

**Figure 5.** The absolute errors for Example 2, with $n = 5$, $n = 10$ and $n = 15$.

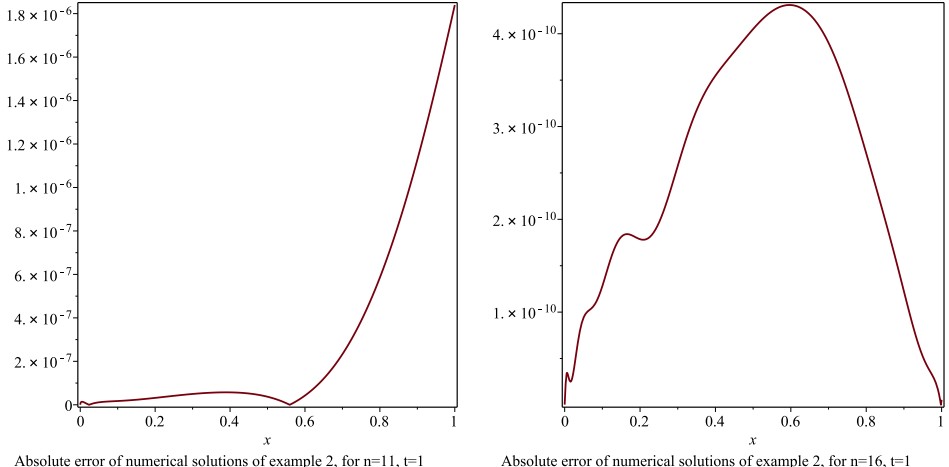

**Figure 6.** The absolute errors for Example 2, with $n = 11$, $n = 16$, and $t = 1$.

By applying the technique in Section 3, for different values of $n$, the FBSSs for the problem are founded. From Tables 12 and 13, we can see obviously that the results obtained by FBSS and the exact solution of the problem are agreement with each other. We compare our results with [42], can see our results are more accurate. The condition numbers for the systems that related to the method for various $n$ are given in Table 14. We can say from Table 14 that the method is insensitive to small variations for $n = 10$.

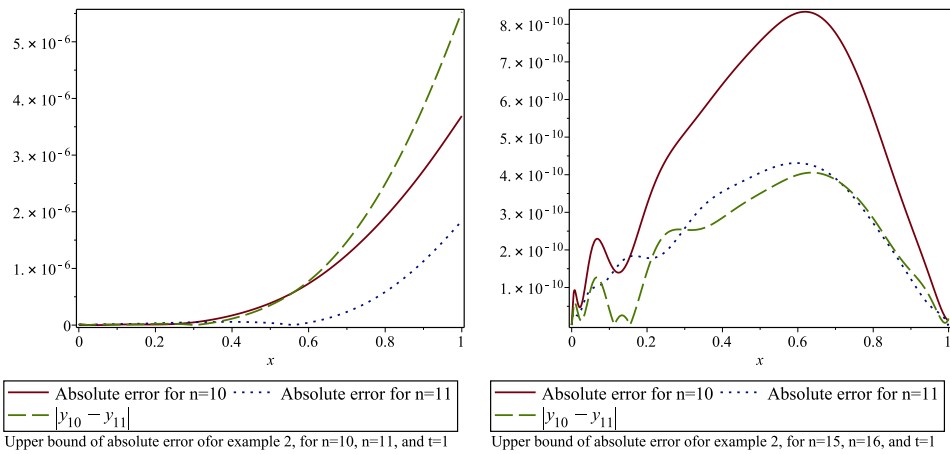

**Figure 7.** Upper bounds of absolute errors for Example 2, with $n = 10$, $n = 11$, and $n = 15$, $n = 16$, for case $t = 1$.

**Table 10.** Error norms $L_2$, $L_\infty$ and CPU for different values of $n$ for Example 2.

| $n$ | $L_2$ | $L_\infty$ | CPU |
|---|---|---|---|
| 3 | $2.294 \times 10^{-1}$ | $1.255 \times 10^{-1}$ | 4.5 |
| 5 | $4.544 \times 10^{-3}$ | $3.461 \times 10^{-3}$ | 7.19 |
| 8 | $5.733 \times 10^{-7}$ | $3.411 \times 10^{-7}$ | 9.5 |
| 10 | $5.538 \times 10^{-8}$ | $3.792 \times 10^{-8}$ | 19.6 |
| 13 | $5.249 \times 10^{-9}$ | $2.287 \times 10^{-9}$ | 62.0 |
| 15 | $1.413 \times 10^{-9}$ | $7.885 \times 10^{-10}$ | 162.3 |
| 18 | $2.916 \times 10^{-10}$ | $1.497 \times 10^{-10}$ | 445 |
| 20 | $8.245 \times 10^{-11}$ | $4.117 \times 10^{-11}$ | 700 |

**Table 11.** $\text{cond}(\bar{W})$ for different values of $n$ for Example 2.

| | $n = 3$ | $n = 5$ | $n = 8$ | $n = 10$ | $n = 13$ | $n = 15$ |
|---|---|---|---|---|---|---|
| $\text{cond}(\bar{W})$ | 105.67 | 851.57 | $72.61 \times 10^3$ | $8.99 \times 10^5$ | $5.03 \times 10^7$ | $9.18 \times 10^8$ |

**Table 12.** Comparisons of the absolute errors for $n = 3, 5, 7$ of Example 3.

| $x$ | [42] $n = 3$ | Our Method $n = 3$ | [42] $n = 5$ | Our Method $n = 5$ | [42] $n = 7$ | Our Method $n = 7$ |
|---|---|---|---|---|---|---|
| 0.1 | $1.63 \times 10^{-5}$ | $2.05 \times 10^{-6}$ | $3.06 \times 10^{-6}$ | $1.20 \times 10^{-7}$ | $2.68 \times 10^{-6}$ | $6.02 \times 10^{-11}$ |
| 0.2 | $4.85 \times 10^{-5}$ | $5.21 \times 10^{-6}$ | $2.08 \times 10^{-6}$ | $1.65 \times 10^{-7}$ | $1.42 \times 10^{-6}$ | $1.50 \times 10^{-10}$ |
| 0.3 | $5.24 \times 10^{-5}$ | $2.70 \times 10^{-6}$ | $2.60 \times 10^{-7}$ | $3.40 \times 10^{-7}$ | $2.06 \times 10^{-6}$ | $1.23 \times 10^{-10}$ |
| 0.4 | $3.67 \times 10^{-5}$ | $9.91 \times 10^{-5}$ | $9.20 \times 10^{-7}$ | $4.09 \times 10^{-7}$ | $4.05 \times 10^{-7}$ | $6.05 \times 10^{-10}$ |
| 0.5 | $1.04 \times 10^{-5}$ | $2.53 \times 10^{-4}$ | $9.70 \times 10^{-7}$ | $6.17 \times 10^{-7}$ | $6.90 \times 10^{-7}$ | $1.49 \times 10^{-10}$ |
| 0.6 | $1.76 \times 10^{-5}$ | $5.04 \times 10^{-4}$ | $1.10 \times 10^{-7}$ | $1.29 \times 10^{-7}$ | $6.00 \times 10^{-7}$ | $2.28 \times 10^{-9}$ |
| 0.7 | $3.85 \times 10^{-5}$ | $8.35 \times 10^{-4}$ | $1.10 \times 10^{-6}$ | $2.60 \times 10^{-6}$ | $7.00 \times 10^{-8}$ | $3.13 \times 10^{-9}$ |
| 0.8 | $4.33 \times 10^{-5}$ | $1.20 \times 10^{-4}$ | $1.92 \times 10^{-6}$ | $4.09 \times 10^{-6}$ | $3.90 \times 10^{-7}$ | $5.34 \times 10^{-9}$ |
| 0.9 | $2.33 \times 10^{-5}$ | $2.19 \times 10^{-4}$ | $2.14 \times 10^{-6}$ | $4.03 \times 10^{-6}$ | $1.84 \times 10^{-6}$ | $8.06 \times 10^{-9}$ |

**Table 13.** Absolute errors for different values of $n$ of Example 3.

| $x$ | $n = 10$ | $n = 13$ | $n = 15$ |
|---|---|---|---|
| 0.1 | $1.92 \times 10^{-15}$ | $5.65 \times 10^{-18}$ | $3.13 \times 10^{-17}$ |
| 0.2 | $1.93 \times 10^{-15}$ | $9.15 \times 10^{-18}$ | $7.82 \times 10^{-18}$ |
| 0.3 | $2.12 \times 10^{-15}$ | $2.14 \times 10^{-17}$ | $4.31 \times 10^{-17}$ |
| 0.4 | $2.64 \times 10^{-14}$ | $6.28 \times 10^{-17}$ | $1.81 \times 10^{-17}$ |
| 0.5 | $6.36 \times 10^{-14}$ | $1.26 \times 10^{-17}$ | $1.49 \times 10^{-16}$ |
| 0.6 | $9.33 \times 10^{-14}$ | $1.34 \times 10^{-17}$ | $3.22 \times 10^{-16}$ |
| 0.7 | $1.65 \times 10^{-13}$ | $4.82 \times 10^{-17}$ | $2.47 \times 10^{-16}$ |
| 0.8 | $2.35 \times 10^{-13}$ | $8.36 \times 10^{-17}$ | $1.78 \times 10^{-16}$ |
| 0.9 | $3.52 \times 10^{-13}$ | $4.98 \times 10^{-17}$ | $6.67 \times 10^{-16}$ |

**Table 14.** $\text{cond}(\bar{W})$ for different values of $n$ for Example 3.

| | $n = 3$ | $n = 5$ | $n = 7$ | $n = 10$ | $n = 13$ | $n = 15$ |
|---|---|---|---|---|---|---|
| $\text{cond}(\bar{W})$ | 47.24 | 308.70 | 2775.84 | 63340.64 | $3.05 \times 10^6$ | $2.92 \times 10^7$ |

## 7. Conclusions

In this paper, we proposed the FBSS to solve the FDE numerically. The method can be easily implemented and is effective. It comprises fractional Bernstein polynomials and the collocation method. First, the fundamental matrix equation is obtained and then it is solved by the Gauss elimination procedure. Applying the initial and boundary conditions, we get the solution for the given $n$ value. Two error estimations, residual correction procedure and estimations are obtained by the difference of the consecutive approximations. We applied the method to some examples. Generally, the results demonstrate that the proposed method achieves better approximation accuracy than some other well-known methods. From the examples, we can bound approximately the absolute error $e_{n,n}$ by $|e_{n,n} - e_{n+1,n+1}|$. On the other hand, one can obtain more accurate results by using the residual correction procedure. Increasing $m$ gives more accurate corrected FBSS. For the stability of the method, we can decide which size $n$ yields more stable approximations by specifying at the number of conditions of the related system. For the examples, the method produces condition numbers approximately as $10^5$ for $n \approx 10$ approximately. Thus, around $n = 10$, we can say that the method is suitable for these examples. The subjects of our future works can be exemplified by applying fractional Bernstein series solution for solving fractional integral differential equations and chaotic fractional order systems.

**Author Contributions:** Conceptualization, M.H.A. and O.I.; methodology, M.H.A. and O.I.; software, M.H.A. and O.I.; validation, I.H. and O.I.; writing—original draft preparation, M.H.A. and O.I.; writing—review and editing, I.H.; supervision, I.H.; funding acquisition, M.H.A. All authors have read and agreed to the published version of the manuscript.

**Funding:** This research was funded by the Abu Dhabi University grant number 18-51-45005.

**Acknowledgments:** The authors acknowledge with thanks, the Department of Mathematics and Statistics at the Abu Dhabi University for making their facilities available for the research.

**Conflicts of Interest:** The authors declare that there is no conflict of interest.

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
