# Peer review of "Fractional Bernstein Series Solution of Fractional Diffusion Equations with Error Estimate"

_axioms, doi:10.3390/axioms10010006_

Round 1

Reviewer 1 Report

The paper is excellent and I can warmly recommend the publication in Axioms.

I suggest the authors to add the following monographs concerning fractional calculus and fractional differential equations in the list of references:

1. E. Bazhlekova, Fractional Evolution Equations in Banach Spaces, Ph.D. Thesis, Eindhoven University of Technology, Eindhoven, 2001.

2. V. Kiryakova, Generalized Fractional Calculus and Applications, Longman Scienti c & Technical, Harlow, 1994, copublished in the United States with John Wiley & Sons, Inc., New York.

3. M. Kostic: Abstract Volterra Integro-Differential Equations, CRC Press, Boca Raton, 2015.

Author Response

First, we thank the respected reviewers for his/her constructive comments, which clearly enhanced the quality of the manuscript.

Referee.1:

The paper is excellent and I can warmly recommend the publication in Axioms.

I suggest the authors to add the following monographs concerning fractional calculus and fractional differential equations in the list of references:

1. E. Bazhlekova, Fractional Evolution Equations in Banach Spaces, Ph.D. Thesis, Eindhoven University of Technology, Eindhoven, 2001.

(We added one reference similar to this reference)

2. V. Kiryakova, Generalized Fractional Calculus and Applications, Longman Scienti

c & Technical, Harlow, 1994, copublished in the United States with John Wiley & Sons, Inc., New York. (Done) (ref. 37)

3. Kostic: Abstract Volterra Integro-Differential Equations, CRC Press, Boca Raton, 2015.

(This reference not related to my paper, but I will add it to my next work, Bernstein method to solve integro differential equation).

Regards

Reviewer 2 Report

This paper studies a numerical method for solving space-fractional partial differential equations, called the fractional Bernstein series solution method. It is based on using fractional Bernstein polynomials, and finite linear combinations thereof, to appropriately approximate the solution function to the FPDE.

This study is heavily oriented towards methodology and examples, with less time devoted to mathematical theorems and proofs, but with many tables and graphs to illustrate the applicability of the studied method. The paper is well presented, professionally written, with a good standard of English. There are just a very few typos, such as "bBy" in the conclusions and "ALshbool" in reference [5], and these are clearly genuine typos rather than due to any lacking English skills.

1. The bibliography contains 25% self-citations (eight references [1,2,3,4,5,18,20,21] among 32 total). Perhaps this is appropriate, given that all eight seem relevant to Bernstein methods; I just wished to remark it.

2. The fractional Bernstein polynomials are defined in equation (3). Please cite a reference for this definition.

3. For interest, it may also be worth mentioning the bivariate Bernstein polynomials and their usage in approximation, see e.g.:
I. Büyükyazici, E. Ibikli, The approximation properties of generalized bernstein polynomials of two variables, Appl. Math. Comp., 156 (2004): 367-380;
M.A. Özarslan, C. Kürt, Bivariate Mittag-Leffler functions arising in the solutions of convolution integral equation with 2D-Laguerre–Konhauser polynomials in the kernel, Appl. Math. Comp., 347 (2019): 631-644.

4. Another related numerical method for fractional differential equations, which could be added to those listed in the introductory review, is the fractional Taylor vector method, according to:
I. Avci, N.I. Mahmudov, Numerical Solutions for Multi-Term Fractional Order Differential Equations with Fractional Taylor Operational Matrix of Fractional Integration, Math., 8(1) (2020): 96.

Author Response

First, we thank the respected reviewers for his/her constructive comments, which clearly enhanced the quality of the manuscript.

Referee.2:

This paper studies a numerical method for solving space-fractional partial differential equations, called the fractional Bernstein series solution method. It is based on using fractional Bernstein polynomials, and finite linear combinations thereof, to appropriately approximate the solution function to the FPDE.

This study is heavily oriented towards methodology and examples, with less time devoted to mathematical theorems and proofs, but with many tables and graphs to illustrate the applicability of the studied method. The paper is well presented, professionally written, with a good standard of English. There are just a very few typos, such as "bBy" in the conclusions and "ALshbool" in reference [5], and these are clearly genuine typos rather than due to any lacking English skills. (Done. Line 229)

1. The bibliography contains 25% self-citations (eight references [1,2,3,4,5,18,20,21] among 32 total). Perhaps this is appropriate, given that all eight seem relevant to Bernstein methods; I just wished to remark it.

 (After reviewer’s comments, I added more than 10 references so my self-citations become 15%) (References before 32, now 42).

 2. The fractional Bernstein polynomials are defined in equation (3). Please cite a reference for this definition.

(Done) (Line 65)

3. For interest, it may also be worth mentioning the bivariate Bernstein polynomials and their usage in approximation, see e.g.:
I. Büyükyazici, E. Ibikli, The approximation properties of generalized bernstein polynomials of two variables, Appl. Math. Comp., 156 (2004): 367-380;

 (Done) (ref. 26).

M.A. Özarslan, C. Kürt, Bivariate Mittag-Leffler functions arising in the solutions of convolution integral equation with 2D-Laguerre–Konhauser polynomials in the kernel, Appl. Math. Comp., 347 (2019): 631-644.

(This reference is not relative for this work; I will add it to my next work Bernstein method to solve integral differential equation)

4. Another related numerical method for fractional differential equations, which could be added to those listed in the introductory review, is the fractional Taylor vector method, according to:
I. Avci, N.I. Mahmudov, Numerical Solutions for Multi-Term Fractional Order Differential Equations with Fractional Taylor Operational Matrix of Fractional Integration,Math., 8(1) (2020):96.

(Done) (ref. 19).

Regards

Reviewer 3 Report

Review: Fractional Bernstein series solution of fractional diffusion equations with error estimate

This paper introduces the fractional Bernstein series solution (FBSS) to solve the fractional diffusion equation.

The paper is written well, and the structure is also good. The theory is given in details. The topic is worth investigating. Publication is recommended considering minor comments.

The introduction needs to be improved. Relevant work should be discussed more. The idea of using a residual to correct or improve solutions can also be found in the work:  Error estimation of the parametric non-intrusive reduced order model using machine learning. Computer Methods in Applied Mechanics and Engineering, 355, 513-534.

In the end of conclusion, what do you mean by bBy in the sentence: ……more stable approximations bBy specifying……

What is your future work and the limitations and disadvantages of your work?  Describe something in the conclusion.

Author Response

First, we thank the respected reviewers for his/her constructive comments, which clearly enhanced the quality of the manuscript.

Referee.3:

Review: Fractional Bernstein series solution of fractional diffusion equations with error estimate

This paper introduces the fractional Bernstein series solution (FBSS) to solve the fractional diffusion equation.

The paper is written well, and the structure is also good. The theory is given in details. The topic is worth investigating. Publication is recommended considering minor comments

1. The introduction needs to be improved. Relevant work should be discussed more. The idea of using a residual to correct or improve solutions can also be found in the work:  

2. Xiao, Error estimation of the parametric non-intrusive reduced order model using machine learning. Computer Methods in Applied Mechanics and Engineering, 355, 513-534.

(Done) (ref. 36).

In the end of conclusion, what do you mean by bBy in the sentence: ……more stable approximations bBy specifying……

(Done, line 229).

What is your future work and the limitations and disadvantages of your work?  Describe something in the conclusion.

 (Done), (line 232-234).

Regards

Reviewer 4 Report

The paper needs major revision. Please see the attached file.

Author Response

Comments on the paper “Fractional Bernstein series solution of fractional diffusion equations with error estimate”
by M.H.T. Alshbool, Osman Isik, I. Hashim
The paper needs major revision.
1. Introduction of the paper is not satisfactory. The authors do not cite the fundamental references on the considered subject.
2. On page 2, the authors write: “Fractional derivatives have been used to model many problems in science”, but cite only several articles. The following books on applications of fractional calculus should be added to References:
Magin, R.L. Fractional Calculus in Bioengineering; Begell House Publishers, Inc.: Redding, CA, USA, 2006.
Mainardi, F. Fractional Calculus and Waves in Linear Viscoelasticity: An Introduction to Mathematical Models; Imperial College Press: London, UK, 2010. Tarasov, V.E. Fractional Dynamics: Applications of Fractional Calculus to Dynamics of Particles, Fields and Media; Springer: Berlin/Heidelberg, Germany, 2010. Povstenko, Y. Fractional Thermoelasticity; Springer: New York, NY, USA, 2015.
3. When the authors discuss operators of fractional order, they cite the less-known article [25]. The following basic books should be cited: Podlubny, I. Fractional Differential Equations; Academic Press: San Diego, CA, USA, 1999.
Kilbas, A.A.; Srivastava, H.M.; Trujillo, J.J. Theory and Applications of Fractional Differential Equations; Elsevier: Amsterdam, The Netherlands, 2006. Povstenko, Y. Linear Fractional Diffusion-Wave Equation for Scientists and Engineers; Birkhäuser: New York, NY, USA, 2015.
4. The authors introduce the Bernstein polynomials without any reference. At least the following book should be cited in this context: Lorentz, G.G. Bernstein Polynomials; Chelsea Publishing Company: New York, NY, USA, 1986.
5. In Introduction, the references should be cited as [20, 21, 26, 31, 32], [3, 5, 11], not as [26, 31, 32, 20, 21], [3, 11, 5].
6. I do not understand why the authors consider Equation (1) in a domain with a boundary conditions at and (see page 2). The Bernstein polynomials are defined for (see Equations (2) and (3)); in all the Examples .
7. A mistake in Equation (27) should be corrected. There should be not
8. The authors refer Example 2 to [29]. In [29] there is no such an Example. This Example was studied in [24].
9. On page 16, Equation (30), the authors write: This is a mistake and misunderstanding. The correct Equation (30) is with the derivative But in this case, Equation (30) is the convection diffusion equation, not fractional convection diffusion equation.
10. The paper should be prepared using the MDPI template.
11. A list of References should be prepared according to the MDPI rules.

Round 2

Reviewer 4 Report

The paper can be published.